# Identification of E3 Ubiquitin Ligase Substrates Using Biotin Ligase-Based Proximity Labeling Approaches

**DOI:** 10.3390/biomedicines13040854

**Published:** 2025-04-02

**Authors:** Koji Matsuhisa, Shinya Sato, Masayuki Kaneko

**Affiliations:** 1Lee Kong Chian School of Medicine, Nanyang Technological University, 50 Nanyang Avenue, Singapore 639798, Singapore; koji.matsuhisa@ntu.edu.sg; 2Department of Pharmacology and Therapeutic Innovation, Nagasaki University Graduate School of Biomedical Sciences, Nagasaki 852-8521, Japan; shin-sato@nagasaki-u.ac.jp

**Keywords:** ubiquitin, ubiquitylation, ubiquitin ligase, biotin ligase, BioID

## Abstract

Ubiquitylation is a post-translational modification originally identified as the first step in protein degradation by the ubiquitin–proteasome system. Ubiquitylation is also known to regulate many cellular processes without degrading the ubiquitylated proteins. Substrate proteins are specifically recognized and ubiquitylated by ubiquitin ligases. It is necessary to identify the substrates for each ubiquitin ligase to understand the physiological and pathological roles of ubiquitylation. Recently, a promiscuous mutant of a biotin ligase derived from *Escherichia coli*, BioID, and its variants have been utilized to analyze protein–protein interaction. In this review, we summarize the current knowledge regarding the molecular mechanisms underlying ubiquitylation, BioID-based approaches for interactome studies, and the application of BirA and its variants for the identification of ubiquitin ligase substrates.

## 1. Background of Protein Ubiquitylation

Protein stability, activity, and localization are regulated by cellular post-translational modifications (PTMs). Ubiquitylation is a PTM defined as the modification of proteins with ubiquitin (Ub), a 76-amino-acid-residue polypeptide. Ubiquitylation is catalyzed by protein complexes composed of a Ub-activating enzyme (E1), a Ub-conjugating enzyme (E2), and a Ub ligase (E3/E3 ligase) (Figure 1) [1,2]. During the first step of ubiquitylation, Ub is activated by the formation of a high-energy thioester bond with E1, coupled with ATP hydrolysis. Ub and the thioester bond are then transferred to E2. E3 ligase recognizes E2 and substrate proteins, and catalyzes the transfer of Ub from E2 to a lysine residue of the substrates. Numerous studies have shown that serine, cysteine, and threonine are also ubiquitylated by the E1-E2-E3 complex in cells [3,4,5]. E3 ligases play important roles in the target specificity of ubiquitylation because of their ability to bind to their substrates. Conjugated Ub undergoes further ubiquitylation [6]. Ub contains seven lysine residues (K6, K11, K27, K29, K33, K48, and K63). All lysine residues and the N-terminal methionine of Ub can be ubiquitylated [7]. As a result of the repeated ubiquitylation of conjugated Ub, a poly-Ub chain is generated in the ubiquitylated protein. Different linkages between Ubs result in distinct chain conformations [8]. In the formation of a poly-Ub chain, all Ubs are assembled with either one type (homogenous) or multiple types (mixed) of linkages. Multiple lysine residues of Ub can be modified simultaneously during poly-Ub formation, generating branched poly-Ub chains. Ubiquitylation is a reversible modification, because Ub conjugated to ubiquitylated proteins can be removed by deubiquitylation [7].

Once a protein is modified with Ub, including mono-Ub and poly-Ub chains, the protein is recognized by Ub-binding proteins, and downstream molecules are activated [10]. The most well-established system activated by ubiquitylation is the degradation of ubiquitylated proteins by the 26S proteasomes [11]. Ubiquitylated proteins are recognized by Ub receptors on the 26S proteasome, such as Rpn10 (yeast) [12], S5a/Rpn10 (mammal) [13], and p54/Rpn10 (fruit fly) [14], and extra-proteasome Ub receptors, such as Dsk2 (yeast) [15], Ddi1 (mammal) [16], and Rad23 (fruit fly) [17]. The proteins are then incorporated into the proteasome and degraded. This Ub–proteasome system contributes to various cellular processes, such as cell cycle regulation [18], the degradation of unfolded/malfolded proteins in the endoplasmic reticulum (ER) [ER-associated degradation (ERAD)] [19], and the activation of NF-κB signaling [20]. In addition, protein ubiquitylation regulates cellular processes without the proteasomal degradation of ubiquitylated proteins [8].

One example is the lysosomal degradation of cell surface and transmembrane proteins localized in organelle membranes [21,22]. Ub acts as a sorting signal for clathrin-mediated internalization. Ubiquitylated plasma membrane proteins are recognized by clathrin-associated proteins Eps15 and Epsin and then sorted into clathrin-coated vesicles. Ubiquitylated transmembrane proteins are also sorted into multivesicular bodies (MVBs) in endosomes. Epidermal growth factor receptor (EGFR), class I major histocompatibility complex-I (MHC-I), and prolactin receptor are incorporated into endosomes via Ub- and clathrin-mediated endocytosis. The endosomal sorting complex required for transport (ESCRT) components, ESCRT-0, I, and II, have Ub-binding domains and sort ubiquitylated transmembrane proteins into MVBs. In particular, the Ub-binding domains of ESCRT-0 are essential for sorting. Ubiquitylated proteins sorted into MVBs are transferred to lysosomes, followed by their degradation.

Mitophagy, a mitochondrion-specific autophagy process, is activated by ubiquitylation to remove damaged mitochondria [23,24]. In depolarized mitochondria, PTEN-induced putative kinase protein 1 (PINK1) accumulates on the outer membrane and phosphorylates Ub, which activates an E3 ligase Parkin and facilitates its recruitment to the mitochondria. Parkin ubiquitylates the outer membrane of damaged mitochondria and Ub-binding autophagy receptors are recruited to promote their capture by autophagosomes.

Ubiquitylation is involved in various cellular processes without proteolysis [3,8]. During the DNA damage response (DDR), ubiquitylated proteins recruit DNA repair factors. For example, ubiquitylated histones H2A and H2A.X recruit TP53-binding protein 1 (53BP) and breast cancer type 1 susceptibility protein (BRCA), which are well-established DDR modulators [25].

Cell division is regulated by ubiquitylation. The E3 ligase anaphase-promoting complex/cyclosome is the main enzyme that targets essential mitotic factors for proteasomal degradation [26]. The anaphase-promoting complex/cyclosome ubiquitylates securin and cyclin B, which are important for regulating sister chromatid separation and mitotic exit, respectively, leading them to the Ub–proteasome pathway for their degradation. Non-proteolytic ubiquitylation plays an important role in cell division. The E3 ligase CUL3 engages the adaptor protein KLHL22, which ubiquitylates polo-like kinase 1 (PLK1), leading to the dissociation of PLK1 from kinetochores and ensuring a timely metaphase to anaphase transition [3,27]. The absence of PLK1 ubiquitylation blocks the dissociation of PLK1 from the kinetochore, resulting in prolonged mitotic arrest and cell death. Ubiquitylation also plays important roles in cell division, organelle homeostasis, and development [3,8].

Protein polyubiquitylation is an important PTM that maintains protein homeostasis and regulates cellular function. Therefore, the identification of substrates for each E3 ligase is important for understanding how cellular processes are maintained and how abnormalities in ubiquitylation cause various diseases.

## 2. Pathological Roles of Protein Ubiquitylation

The dysfunction and aberrant activation of the ubiquitylation machinery can cause diseases. Indeed, numerous studies have implicated abnormalities in ubiquitylation in the pathogenesis of various diseases, such as neurodegenerative diseases [28,29], neurodevelopmental disorders [30,31,32], cancers [7,33], immune diseases [34], and metabolic diseases [35]. In this section, we briefly describe current knowledge regarding the role of ubiquitylation in the pathogenesis of neurodegenerative diseases and cancer.

### 2.1. Ubiquitylation and Neurodegenerative Diseases

The abnormal accumulation of neurotoxic proteins, such as amyloid-β, tau, α-synuclein, and mutant huntingtin, is a hallmark of neurodegenerative diseases [36,37,38,39]. It has been reported that those neurotoxic proteins are directly or indirectly degraded by ubiquitylation. Tau, α-synuclein, and mutant huntingtin are the causative factors of Alzheimer’s disease, Parkinson’s disease, and Huntington’s disease, respectively. They can be ubiquitylated by the carboxyl terminus of Hsp70-interacting protein (CHIP), an E3 ligase, followed by degradation through the Ub–proteasome system [40,41,42,43]. CHIP can also reduce the production of amyloid-β, another causative factor of Alzheimer’s disease, by reducing the cleavage of amyloid-β precursor protein (APP) via the downregulation of β-site app-cleaving enzyme 1 (BACE1) [44].

Mitochondrial dysfunction is involved in the pathogenesis of neurodegenerative diseases. Mutations in Parkin are a common cause of familial Parkinson’s disease [45]. In Parkin-knockout mice, damaged mitochondria accumulated in the dopaminergic neurons of 110-week-old mice, and dopaminergic neural loss was observed at 120 weeks [46]. Several studies have suggested that Parkin-mediated mitophagy is related to the pathogenesis of Alzheimer’s disease [29,47]. Pathological tau protein inhibits Parkin recruitment to impaired mitochondria, thereby disrupting mitochondrial quality control [48].

Additionally, ubiquitylation is involved in neuroinflammation [49], excitotoxicity [50], ER stress [42], calcium overload [51], and cell cycle abnormalities [52]. Therefore, ubiquitylation is an attractive target for treating neurodegenerative diseases.

### 2.2. Ubiquitylation and Neurodevelopmental Disorders

Neurodevelopmental disorders are defined as “a group of conditions with onset in the developmental period, inducing deficits that produce impairments of functioning” in the New Diagnostic and Statistical Manual of Mental Disorders, 5th Edition (DSM-5) [53]. Approximately 40% of neurodevelopmental disorders are estimated to be caused by monogenic conditions, predominantly lesions in a single gene [31]. To date, many variants of E3 ligase-encoding genes have been identified in patients with neurodevelopmental disorders and proposed as causative factors for these disorders [31,54].

The *UBE3A* gene encoding the E6-AP HECT-type E3 ubiquitin ligase and loss-of-function mutation in the *UBE3A* has been shown to be a causative factor of Angelman syndrome, a neurodevelopmental disorder [31,55,56,57]. Intracellular localization and biochemical analyses revealed that 55% of the variants found in patients with Angelman syndrome induced the mislocalization of UBE3A, and 29% of the variants induced a loss of E3 ligase activity [58]. Although the critical substrates of UBE3A are unknown, Arc has been identified as a substrate of UBE3A in neuronal cells [59]. Arc regulates the trafficking of alpha-amino-3-hydroxy-5-methyl-4-isoxazole-propionate-type glutamate receptors at synapses [60,61,62]. UBE3A ubiquitylates Arc and induces its degradation by the Ub–proteasome system; the depletion of UBE3A increases Arc, followed by a reduction in the cell surface expression of the alpha-amino-3-hydroxy-5-methyl-4-isoxazole-propionate receptor [58].

Genetic variants of the HECT E3 ligase NEDD4 subfamily have been proposed as causative factors of neurodevelopmental disorders [32]. The NEDD4 subfamily consists of nine E3 ligases: NEDD4-1, NEDD4-2, ITCH, WWP1, WWP2, SMURF1, SMURF2, NEDL1, and NEDL2. These HECT E3 ligases play crucial roles in neurodevelopment [32]: NEDD4-1, SMURF1/2, WWP2, and ITCH1 in neuronal proliferation [63,64,65,66,67,68]; NEDD4-1/2, SMURF1, WWP1/2, and HECW2 in the migration and differentiation of neuronal cells [69,70,71,72,73,74,75,76,77,78,79,80]; and NEDD4-1/2 in the connections among neurons [81,82,83,84,85]. Genetic variants of NEDD4 have also been identified in patients with neurodevelopmental disorders. Genetic variants of *NEDD4-2* have been identified in patients with periventricular nodular heterotopia, polymicrogyria, macrocephaly, cleft palate, and syndactyly [71]. Several studies have identified de novo mutations in the HECW gene in patients with neurodevelopmental diseases, including epilepsy, intellectual deficiency, and macrocephaly [77,78,79,80].

To date, pathogenic variants have been identified in at least 53 genes encoding E3 ligases in patients with neurodevelopmental disorders [54]. Further studies regarding the pathological relevance of E3 ligases in these disorders would be helpful for developing novel therapeutic approaches for these disorders.

### 2.3. Ubiquitylation and Cancer

Cancer cells modulate various cellular processes to actively and infinitely proliferate and evade anticancer systems, such as apoptosis and immunity, in the body. Cancer cells hijack ubiquitylation to induce this modulation [7].

Numerous studies have shown that the dysregulation of ubiquitylation is involved in abnormal proliferation [7]. The E3 ligase tumor necrosis factor receptor-associated factor 4 (TRAF4) is highly expressed in metastatic prostate cancer [86]. Recent studies have shown that TRAF4 ubiquitylates the androgen receptor, promoting proliferation via the activation of the protein kinase A pathway [87]. Abnormal E3 ligase activity also supports the inactivation of tumor suppressors such as p53 and retinoblastoma protein (RB) [88,89]. p53 plays pivotal roles in cell cycle regulation, apoptosis in response to DNA damage, and DNA repair [90]. p53 degradation is facilitated by its ubiquitylation by the E3 ligases mouse double minute 2 homolog (MDM2) and tripartite motif-containing 28 (TRIM28) in various cancer cells [88,89]. TRIM28 also ubiquitylates phosphorylated RB, promoting its degradation [89].

Hijacking ubiquitylation also contributes to the invasive and metastatic abilities of tumors. F-Box and WD repeat domain containing 2 (FBXW2) is an E3 ligase that inhibits cancer migration, invasion, and metastasis by promoting the degradation of oncogenic proteins, such as S-phase kinase associated protein 2 (SKP2) and β-catenin [91,92]. It has also been reported that an E3 ligase β-transducin repeat-containing protein 1 (β-TrCP1) ubiquitylates FBXW2, promoting its degradation in lung cancer cells [92].

Numerous reports have suggested that ubiquitylation is also involved in the suppression of cancer cell death, including apoptosis and necroptosis; replicative immortality; angiogenesis; genome instability and mutation; tumor-promoting inflammation; reprogramming energy metabolism; evading immune destruction; unlocking phenotypic plasticity; epigenetic reprogramming; the generation of a polymorphic microbiome; and cellular senescence [7]. Therefore, many drugs targeting ubiquitylation have been developed to establish novel and effective cancer treatments.

## 3. Ubiquitylation as a Therapeutic Target

Protein ubiquitylation is an attractive target for the development of medicines. Various therapeutic approaches using ubiquitylation have been proposed.

Thalidomide is a sedative with teratogenic side effects; however, its derivatives, including lenalidomide and pomalidomide, are also relatively safe immunomodulatory drugs (IMiDs) for patients with cancer, except for pregnant women [93]. IMiDs have been shown to interact with cereblon (CRBN), an adapter protein of E3 ligase, recruiting proteins such as the Ikaros family zinc finger protein 1 (IKZF1) and IKZF3, which play central roles in the biology of B and T cells, as neo-substrates [94,95,96,97,98,99,100]. This interaction facilitates the proteasome-mediated degradation of IKZF1 and IKZF3. This unique characteristic of IMiDs is related to their anti-multiple-myeloma activities.

CC-885 was identified as a CRBN ligand with unique activities from a library of thalidomide analogs [101]. CC-885 binds to CRBN and GSPT1 (eRF3a), a translation termination factor that induces ubiquitination and degradation by the Ub–proteasome system, as well as IKZF1 and IKZF3. CC-885 exhibits anti-tumor effects through the degradation of GSPT1.

Indisulam (E7070) is a sulphonamide anti-tumor drug [93]. Indisulam disrupts and reduces the levels of cyclin A, cyclin B, CDK2, and CDC2 through mechanisms that depend on p21 and p53, affecting multiple checkpoints during the G1 and G2 phases of the cell cycle [102]. A study on indisulam resistance, which is caused by mutations in RBM39, an RNA-binding nuclear protein, revealed that indisulam binds to both RBM39 and a component of the CUL4 E3 ligase complex, DCAF15, inducing polyubiquitination and the degradation of RBM39 [103]. The degradation of RBM39 induced by indisulam treatment leads to aberrant pre-mRNA splicing, which, in turn, induces cell death in cancer cell lines that highly express RBM39.

Recently, a promising therapeutic approach, proteolysis-targeting chimera (PROTAC) technology, was introduced to promote the degradation of target proteins via the Ub–proteasome system [104]. PROTAC is a dual-binder comprising an E3 ligase-binding ligand and a protein of interest (POI)-binding ligand. PROTAC recruits a POI to its target E3 ligase, followed by the ubiquitylation and degradation of the POI.

Vepdegestrant (ARV-471) is a PROTAC composed of a lasofoxifene-based estrogen receptor-targeting moiety linked to a derivative of lenalidomide, a clinical IMiD that binds to CRBN [105]. Vepdegestrant induces the degradation of the estrogen receptor through the Ub–proteasome system via polyubiquitylation by CRBN-containing E3 ligase complexes. A preclinical study using MCF7 orthotopic xenograft models and a patient-derived xenograft breast cancer model has demonstrated that vepdegestrant has potent anti-tumor activity against estrogen receptor-positive tumors [105]. A phase III clinical trial is currently in progress [106].

Dual-binding molecules targeting E3 ligases and pathogenic proteins/disease-specific proteins are promising therapeutic drugs for treating intractable cancers and various diseases that are induced by pathogenic proteins or expressed disease-specific proteins.

## 4. Biotin-Ligase-Based Protein–Protein Interactor Screening System

It is important to identify interactions between proteins to understand their physiological and pathological roles in the body. Traditionally, scientists have sought to find interactors of POIs by collecting protein complexes using affinity-based approaches, such as coimmunoprecipitation and pull-down, or yeast two-hybrid systems. However, detecting transient and weak interactions using affinity-based approaches is difficult. Yeast two-hybrid systems cannot completely reproduce the intrinsic nature of POIs because various conditions, such as the environment, associated proteins, and post-translational modifications of target proteins, are lost. In the past decade, proximity labeling using a promiscuous mutant biotin ligase, BioID, has been used to analyze protein–protein interactions (PPIs) and identify interacting proteins [107]. In this section, we provide basic information about BioID and BioID-based interactor screening and introduce improved BioID proteins and other biotin ligases developed for interactome studies.

### 4.1. Basic Information About BioID and BioID-Based Interactor Screening Systems

BioID contains a BirA mutant (BirA R118G) [108]. BirA is a 35 kDa DNA-binding protein ligase derived from *E. coli* that regulates the biotinylation of a subunit of acetyl-CoA carboxylase [109]. BirA produces and holds the reactive biotin molecule, biotinoyl-5′-AMP (bioAMP); recognizes a 14-amino-acid-residue peptide; and biotinylates using bioAMP [110,111]. In contrast, the affinity between BioID and bioAMP is two orders of magnitude lower than that between wild-type BirA and bioAMP [112]. Thus, BioID promiscuously biotinylates proteins in a proximity-dependent manner. In 2012, Roux et al. developed a BioID-based proximity labeling system [108]. In this system, a POI is tagged with BioID, and the tagged construct is overexpressed in target cells (Figure 2). The tagged construct biotinylates interacting proteins and neighboring proteins that exist within approximately 10 nm of BioID [113]. Biotinylated proteins are collected using avidin beads, and interacting and neighboring proteins are analyzed using mass spectrometry. Kyle et al. showed that interacting and neighboring proteins could be screened using BioID-tagged Lamin A, a model construct [108]. In their study, they detected FAM169A (KIAA0888) as a novel nuclear envelope (NE) constituent, as well as known NE components, such as β and γ isoforms of lamina-associated polypeptide 2 (LAP2) and LAP1. The bioID-based approach is considered helpful for analyzing weak and/or transient interactions. The affinity-based approach requires the maintenance of PPIs until the protein complexes are collected. In general, PPIs, especially those with weak interactions, are disrupted by detergents. In contrast, BioID biotinylates interactors before cell lysis; thus, protein interactions do not need to be maintained. Therefore, BioID-based approaches are preferred for analyzing weak and transient interactions.

### 4.2. Improved BioID and Other Biotin Ligases for Proximity Ligation

Various BioID derivatives and other biotin ligases have been developed to overcome the limitations of BioID (Table 1). Such derivatives and biotin ligases would be helpful for investigating the interactions between substrates/interactors and E3 ligases. In this subsection, we summarize improved BioIDs and other biotin ligases for proximity ligation, although some have not been used to identify substrates and interactors for E3 ligases.

Although BioID is an attractive tool for interactome studies, it has several limitations. First, more than 16 h are required to biotinylate the interacting proteins and neighboring proteins with BioID [108]. Therefore, BioID is unsuitable for time-sensitive experiments. Second, BioID is relatively large (35 kDa), which disrupts the intrinsic localization of the POI in some cases [114]. Several biotin ligases have been developed to overcome these limitations and to increase the utility of BioID-based methods.

BioID2 is a promiscuous mutant of a biotin ligase derived from *A. aeolicus*, with a smaller size (233 amino acids) than BioID (321 amino acids) [114]. The SAD1/UNC84 domain protein (SUN2) is an NE protein that is sensitive to ER mislocalization upon the fusion of bulky motifs to its N-terminus [124,125]. BioID-tagged SUN2 partially localized to the ER; however, BioID2-tagged SUN2 enabled more appropriate targeting of SUN2 to the NE than achieved using BioID [114]. Using BioGRID, we confirmed that, in the top ten hits identified using BioID2-fused nucleoporin component Nup43, the interaction of two of the candidates with Nup43 has been shown by other methods [126,127]. BioID2 may be useful when the localization of a POI is disturbed by tagging it with BioID.

TurboID is a potent BioID [115] mutant that was developed by combining random mutagenesis and yeast surface display. MiniTurbo, which has a smaller molecular weight (28 kDa) compared to BioID and TurboID (35 kDa), was prepared using the same approach in the same study. TurboID produced almost as much biotinylated product in 10 min as BioID produced in 18 h. Hence, TurboID may be preferable for time-sensitive studies. In contrast, decreased fruit fly viability and size were observed when TurboID was expressed ubiquitously and constitutively. Developmental delay was also evident in *C. elegans,* in which TurboID is constitutively expressed. These toxicities may be caused by the consumption of endogenous biotin or the inhibition of endogenous protein activities by biotinylation. MiniTurbo showed 1.5 to 2 times less biotinylation activity than TurboID, but miniTurbo was also able to biotinylate neighboring proteins within 10 min.

AirID was designed using an ancestral enzyme reconstruction algorithm and a large genome dataset [116]. Protein biotinylation in the AirID-expressing cells treated with biotin for 3 h was higher than that in the BioID-expressing cells treated with biotin for 16 h. An in vitro bioAMP production assay using radioisotope-labeled ATP revealed that AirID has a higher bioAMP formation activity than TurboID, although AirID biotinylation activity was slightly lower than that of Turbo ID. Unlike TurboID, AirID did not show cytotoxicity under 50 μM biotin supplementation. This suggests that AirID is more suitable for experiments that require long-term biotin treatment and constitutive expression.

A modified biotin ligase derived from *Bacillus subtilis* (BASU) was used as a BioID-like biotin ligase [117]. After 1 min of labeling, BASU achieved >30-fold higher enrichment of interactor protein candidates compared to 18 h of labeling using BioID.

MicroID2 and UltraID are small biotin ligases (180 and 170 amino acids, respectively) that are truncated mutants of BioID2 with some substitution mutations [118,119]. MicroID2 shows slightly less biotinylation activity than TurboID and considerably stronger activity than miniTurbo and BioID2 after 1 h of biotin treatment [118]. UltraID shows a biotinylation activity similar to that of TurboID after both 10 min and 1 h of biotin treatment, suggesting that UltraID may be more potent than MicroID2 [119].

BioID can be separated into two inactive fragments that can be reconstituted as active BioID molecules. This approach is known as split-BioID [120,121]. The N-terminal- and C-terminal fragments of BioID show biotinylation activity upon interaction with each other. Similarly to split-BioID, split-TurboID and split-AirID systems have also been established [122,123]. These reconstitutive BioID variants will be helpful in investigating the interactomes of protein dimers.

## 5. Identification of Ub E3 Ligase Substrates Using Biotin-Ligase-Based Proximity Labeling Approaches

Although many substrates for E3 ligases have been identified thus far, it is difficult to identify novel substrates for E3 ligases using conventional coimmunoprecipitation and pull-down approaches, because the interaction between the Ub ligase complex and its substrate is transient. BioID-based approaches have been established as tools for investigating PPIs. BioID-based approaches may also be helpful for identifying E3 ligase substrates. Various proteins have been identified as substrates for E3 ligases using BioID and its variants.

β-TrCP1 (BTRC) and β-TrCP2 (FBXW11) are E3 ligases that target substrates for 26S proteasome-mediated degradation [128]. BioID-based screening using BioID-tagged β-TrCP1 and β-TrCP2 identified 12 proteins, SUN2, TRIM9, CREBRF, CTNNA1, CNOT10, TNRC6B, KANK2, STK3 (MST2), MAP1S, MKLN1, RASSF3, and PPP1R15B, as substrates of the E3 ligase [129]. Subsequent experiments revealed that the β-TrCP-dependent degradation of these substrates is critical for nuclear membrane integrity, P-body abundance, and the regulation of eukaryotic initiation factor 2α (eIF2α) phosphorylation [129]. Our protein–protein interaction network analysis using the software Cytoscape (https://cytoscape.org, version 3.10.3) (accessed on 21 February 2025) and a database String (using a plug-in for Cytoscape, stringApp) shows that RASSF3 and PPP1R15B interact with β-TrCP1 and β-TrCP2, respectively, TRIM9 and SUN2 directly interact with both β-TrCP1 and β-TrCP2, and STK3 and MAP1S indirectly interact with β-TrCP1, supporting their findings (Figure 3).

Ring finger protein 183 (RNF183) is an E3 ligase specifically expressed in the kidney, especially in the renal medulla [130]. Under normal conditions, the renal medulla is hypertonic. Using BioID-tagged RNF183, Na, K-ATPase α1 subunit was identified as a substrate protein of RNF183 [130]. Intriguingly, RNF183 ubiquitinated only the Na, K-ATPase β1 subunit and not the α1 subunit. After the ubiquitylation of the β1 subunit by RNF183, the Na, K-ATPase complex was sorted to the lysosome, and then degraded.

BioID-based screening using AirID-tagged CRBN revealed that zinc finger MYM-type containing 2 (ZMYM2) and ZMYM2-FGFR1 fusion proteins are also neo-substrates for CRBN [131]. This AirID-based approach can also be applied to other molecules that recruit neo-substrates to a specific E3 ligase [131].

BioID2-based proximity labeling can also be used to investigate the toxic effects of IMiDs [132]. The screening of substrates/neo-substrates for CRBN/IMiD complexes using BioID2-fused CRBN identified non-muscle myosin heavy chain IIA (MYH9) as the neo-substrate. Although the protein level of MYH9 was not changed by treatment with IMiDs, ubiquitylated MYH9 levels increased in the presence of IMiDs. This study suggests the potential of BioID-based proximity labeling to identify neo-substrates of IMiDs and predict their side effects.

Wild-type BirA has also been utilized to identify E3 ligase substrates. Unlike promiscuous biotin ligases such as BioID, wild-type BirA does not diffuse bioAMP to the surrounding environment. In this approach, known as BioE3, E-STAB, or Ub-POD, Ub tagged with a biotin acceptor peptide such as AviTag is used as an acceptor for biotinylation by BirA-tagged E3 ligases [133,134,135]. Since BirA biotinylates its target sequence in a specific manner, ubiquitylated proteins should be preferentially identified using this approach. Additionally, these approaches used modified biotin acceptor peptides with low affinity for BirA to reduce the background biotinylation induced by the interaction between the acceptor peptides and BirA. The wild-type BirA-based approaches have been used to identify various candidate substrates for E3 ligases, such as many components of the replication fork and proteins with helicase activity (as substrates for RNF4) [133], ESCRT protein signal transducing adapter molecule (STAM) [135], hepatocyte growth factor-regulated tyrosine kinase substrate (HGS) [135], and selective autophagy receptor Toll-interacting protein (as substrates for TRAF6), annexin (as a substrate for CHIP) [135], vinculin (VCL) and eukaryotic translation termination factor 1 (ETF1) (as neo-substrates for CRBN and an IMiD CC-885) [134], several kinases (as neo-substrates for CRBN and multikinase degraders SK-3-91, DB0646, SB1-G-187, and WH-10417-099) [134], SMRT/NcoR, MIER, CoREST, MiDAC, SIN3, and NuRD complexes [as neo-substrates for VHL and a dacinostat-based multi-histone deacetylase (HDAC) degrader XY-07-187] [134], and HIF1a (as a substrate for VHL) [134].

## 6. Conclusions

Several E3 ligase substrates have been identified using BioID-based approaches. BioID and its variants can also be used for in vivo interactome studies in animals, such as flies [136], *C. elegans* [137], and mice [138]. Thus, BioID and its variants may be useful for identifying novel pathogenic substrates of E3 ligases in animal disease models.

If PROTAC binds to undesired proteins, off-target effects may induce severe side effects. It may be possible to predict such side effects by identifying the substrates for each PROTAC using BioID-based approaches, such as neo-substrate screening for IMiDs.

Many E3 ligases are promising targets for the treatment of several diseases including neurodegenerative diseases and cancer [7,29]. BioID and its variants would be helpful in unveiling the proteins ubiquitylated by each E3 ligase, leading to the development of novel and effective treatments for various diseases.

Although biotin ligase-based proximity labeling is helpful for identifying substrates for E3 ligases, validation using other methods, such as immunoprecipitation using cells expressing both substrate candidates and the E3 ligase of interest, is required to confirm that the identified candidates are true substrates. Defects of the ubiquitylation of the candidates should also be observed in E3 ligase of interest-knockout/knockdown cells.

In most studies, biotin ligase-based proximity labeling has been performed by overexpressing the biotin ligase-fused POI. However, excess exogenous expression induces the mislocalization and aberrant binding/biotinylation of the fusion protein, resulting in artifact detection [139]. To avoid such false candidates, the endogenous tagging of biotin ligases to genes encoding POI using genome editing would be helpful. A study using biotin ligase-tagged ubiquitin-specific peptidase 46 knock-in (KI) cells has identified several centrosomal proteins as interactors of ubiquitin-specific peptidase 46, suggesting the potential of biotin ligase-E3 ligase in KI cells to identify its substrates. In vivo approaches using biotin ligase-fused E3 ligase of interest in KI animals would also be beneficial, especially for screening E3 ligase substrates that are ubiquitylated under specific physiological/pathological conditions.

The ubiquitylation of substrates by E3 ligases is regulated by regulatory proteins such as kinases and other E3 ligases via post-translational modifications [140]. Understanding the regulation of ubiquitylation by each E3 ligase is important for the development of novel therapeutic approaches. Biotin ligase-based proximity labeling approaches may also be useful for identifying such regulatory proteins for the E3 ligase of interest.

In addition to E3 ligase substrates/interactors, other aspects of ubiquitylation, such as the ubiquitome, have been analyzed using biotin ligases [141,142,143,144]. Several questions regarding ubiquitylation could be resolved using biotin ligase-based proximity labeling approaches.

## Figures and Tables

**Figure 1 biomedicines-13-00854-f001:**
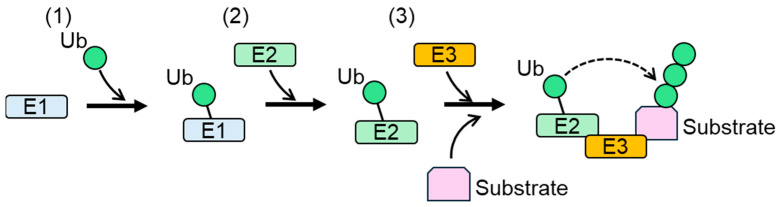
Protein ubiquitylation by E1, E2, and E3 [9]. (**1**) Ub is activated and crosslinked to E1. (**2**) Activated Ub is transferred from E1 to E2. (**3**) E3 binds both Ub-linked E2 and a substrate, and then catalyzes the transfer of Ub from E2 to a lysine residue of the substrate or Ub conjugated to the substrate. Ub: ubiquitin.

**Figure 2 biomedicines-13-00854-f002:**
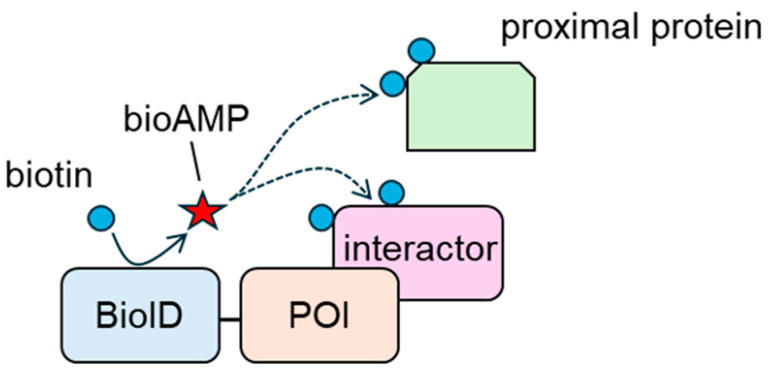
Biotinylation of interacting proteins and proximal proteins of the protein of interest (POI) by BioID. bioAMP: biotinoyl-5′-AMP.

**Figure 3 biomedicines-13-00854-f003:**
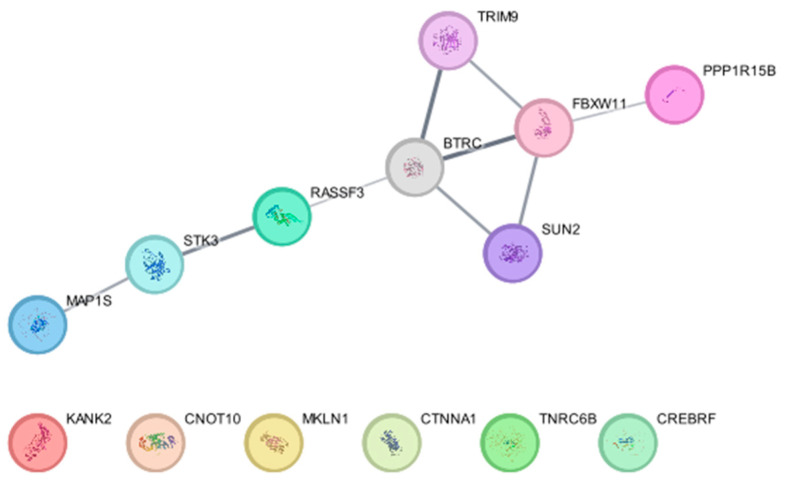
Protein–protein interaction of β-TrCP1 (BTRC) and β-TrCP2 (FBXW11) with their substrates identified in Coyaud, E. et al. [129].

**Table 1 biomedicines-13-00854-t001:** BioID and other biotin ligases used for proximity ligation. The biotin incubation time used for proteomics analysis in the indicated studies is shown as the incubation time for biotinylation. (N) and (C) indicate the N- and C-terminal fragments, respectively.

Biotin Ligase	Amino Acid Length	Incubation Time for Biotinylation	Reference
BioID	321	<16 h	[108]
BioID2	233	16 h	[114]
TurboID	321	10 min	[115]
Miniturbo	254	N.D.	[115]
AirID	317	6 h	[116]
BASU	325	18 h	[117]
MicroID2	180	3 h	[118]
UltraID	170	10 min	[119]
Split-BioID	140 (N), 181 (C)	16 h	[120]
	256 (N), 65 (C)	24 h	[121]
Split-TurboID	73 (N)	4 h	[122]
	248 (C)		
Split-AirID	98 (N)	<24 h	[123]
	245 (C)		

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
