# Peer review of "Identification of E3 Ubiquitin Ligase Substrates Using Biotin Ligase-Based Proximity Labeling Approaches"

_biomedicines, 2025, doi:10.3390/biomedicines13040854_

Round 1

Reviewer 1 Report

Comments and Suggestions for Authors

In this review, Matsuhisa and colleagues very briefly introduce the ubiquitination machinery and provide an overview of the BioID-based proximity labeling approach, which is useful for identifying E3 ubiquitin ligase substrates (and interactors?). Identifying the specific substrates (and interactors?) of E3 ligases is exceedingly difficult; therefore, any method designed to facilitate this process would be beneficial for future research on the ubiquitin-proteasome system (UPS).
The review is relatively short, and while appropriately written, I believe that a short review or mini-review format would be more suitable. Moreover, if the current title is retained, a more comprehensive introduction and additional results regarding the BioID-based E3 substrate screen should be included. That said, the review provides a well-summarized discussion with 82 references, making it a helpful guide for those interested in the UPS and its substrates.

Minor concerns:
    Title clarity: The title is somewhat unclear. Instead of "biotin-ligase-based approaches," it should be "BioID-based proximity labeling approaches" or "biotin-ligase-based proximity labeling approaches," as the emphasis is on proximity labeling.
    Line 38-39: "All lysine residues and the N-terminal methionine of Ub can be ubiquitylated." Please provide a citation, as this is not trivial.
    Line 45-46: "Ubiquitylation is a reversible modification because Ub conjugated to ubiquitylated proteins can be removed by deubiquitylation." Please provide citations on deubiquitinases (DUBs), as they are crucial components of the UPS and might also function as substrates or interactors of E3 ligases.
    Line 53: "by 26S proteasome" should be "by the 26S proteasome."
    Line 53-54: "Ubiquitylated proteins are recognized by Ub receptors on the 26S proteasome." It is surprising that major elements of the UPS, such as Ub receptors, are not mentioned or cited. I believe that at least the fundamental studies identifying Rpn10 in yeast, S5a/Rpn10 in mammalian cells, and p54/Rpn10 in fruit flies should be cited. Additionally, extra-proteasomal receptors such as Dsk2, Ddi1, and Rad23 in yeast, humans, and fruit flies should also be referenced.
    Line 209: "fly" should be either "fruit fly" or "Drosophila melanogaster."
    Line 239: The subtitle is also misleading, like the main title. It should be "BioID-based proximity labeling approaches" or similar.
    Line 284: PROTACs is not a "new therapeutic approach," as it was developed in the early 2000s.
    Discussion suggestion: In addition to identifying E3 substrates, this approach can also uncover regulators and other interactors. I suggest including a brief discussion on these aspects as well.

Author Response

1) Title clarity: The title is somewhat unclear. Instead of "biotin-ligase-based approaches," it should be "BioID-based proximity labeling approaches" or "biotin-ligase-based proximity labeling approaches," as the emphasis is on proximity labeling.

   We appreciate your suggestion. We changed the title from “Identification of E3 ubiquitin ligase substrates using biotin ligase-based approaches” to “Identification of E3 Ubiquitin Ligase Substrates Using Biotin Ligase-based Proximity Labeling Approaches” (page 1, line 2 – line 3).

2) Line 38-39: "All lysine residues and the N-terminal methionine of Ub can be ubiquitylated." Please provide a citation, as this is not trivial.

   We apologize for the lack of proper citation. We added a reference (Liu et al., Mol. Cancer 2024) to the sentence which reviewer #1 pointed out as a citation (page 1, line 38).

3) Line 45-46: "Ubiquitylation is a reversible modification because Ub conjugated to ubiquitylated proteins can be removed by deubiquitylation." Please provide citations on deubiquitinases (DUBs), as they are crucial components of the UPS and might also function as substrates or interactors of E3 ligases.

   As reviewer #1 suggested, we added a reference (Liu et al., Mol. Cancer 2024) to the sentence which the reviewer1 pointed out as a citation (page 2, line 44).

4) Line 53: "by 26S proteasome" should be "by the 26S proteasome."

   As reviewer #1 suggested, we corrected “by 26S proteasome” to “by the 26S proteasome” (page 2, line 55).

5) Line 53-54: "Ubiquitylated proteins are recognized by Ub receptors on the 26S proteasome." It is surprising that major elements of the UPS, such as Ub receptors, are not mentioned or cited. I believe that at least the fundamental studies identifying Rpn10 in yeast, S5a/Rpn10 in mammalian cells, and p54/Rpn10 in fruit flies should be cited. Additionally, extra-proteasomal receptors such as Dsk2, Ddi1, and Rad23 in yeast, humans, and fruit flies should also be referenced.

   As reviewer #1 pointed out, we added some citations and mentioned some Ub receptors in the revised manuscript as below (page 2, line 54 – page 2, line 57) .

Ubiquitylated proteins are recognized by Ub receptors on the 26S proteasome, such as Rpn10 (yeast)[12], S5a/Rpn10 (mammal)[13], p54/Rpn10 (fruit fly)[14], and extra-proteasome Ub receptors, such as Dsk2 (yeast)[15], Ddi1 (mammal)[16], and Rad23 (fruit fly)[17].

6) Line 209: "fly" should be either "fruit fly" or "Drosophila melanogaster."

   As pointed out, we corrected “fly” to “fruit fly” the main text in the revised manuscript (page 7, line 321) .

7) Line 239: The subtitle is also misleading, like the main title. It should be "BioID-based proximity labeling approaches" or similar.

   As review #1 suggested, we changed the subtitle from “Identification of Ub E3 ligase substrates using a biotin-ligase-based screening system” to “Identification of Ub E3 ligase substrates using a biotin-ligase-based proximity labeling approaches” (page 8, line 351-352).

8) Line 284: PROTACs is not a "new therapeutic approach," as it was developed in the early 2000s.

   As reviewer #1 and #3 pointed out, we changed the sentence as below and moved it to section 3 (page 5, line 220 – page 5, line 222).

Recently, a promising therapeutic approach, proteolysis-targeting chimera (PROTAC) technology, was introduced to promote the degradation of target proteins via the Ub-proteasome system[104].

9) Discussion suggestion: In addition to identifying E3 substrates, this approach can also uncover regulators and other interactors. I suggest including a brief discussion on these aspects as well.

   As suggested, we added a brief discussion on identification of regulators and other interactors of E3 ligases as below (page 10, line 437 – page 10, line 441).

   The ubiquitylation of substrates by E3 ligases is regulated by regulatory proteins such as kinases and other E3 ligases via post-translational modifications[140]. Understanding the regulation of ubiquitylation by each E3 ligase is important for the development of novel therapeutic approaches. Biotin ligase-based proximity labeling approaches may also be useful for identifying such regulatory proteins for the E3 ligase of interest.

Reviewer 2 Report

Comments and Suggestions for Authors

The review prepared by Matsuhisa, Sato and Kaneko entitled “Identification of E3 ubiquitin ligase substrates using biotin ligase-based approaches” covers at length the topics of the ubiquitin cycle and the biotin-based tools available for studies in cells and organisms, but spends very little time/text discussing the exact topic mentioned in the title.   It is recommended that authors revise carefully, to streamline the introductory materials and spend more time reviewing, comparing and contrasting and accurately reporting on the few papers that use biotin-based methods to identify putative E3 ligase substrates.  A section on “future challenges” in this area would also be useful. Specific comments: 11-12> “Ubiquitylation is a post-translational modification that was originally identified as a molecular probe that leads labeled proteins…”  What is meant by “molecular probe”? 39-40> “The 3D structure of the Ub-Ub molecule is differentiated by the position of the modulated lysine residue in the N-terminal Ub…”  This sentence is not clear and should be re-written; cannot understand what authors are trying to explain. 72-77> This paragraph seems out of place, but “parkin” is mentioned, however not highlighted as an “E3 ubiquitin ligase”. 84, 85> The concept of this sentence/ paragraph is repeated in the next one (line 88) The title of the work is “Identification of E3 ligase substrates using biotin ligase-based approaches”. I feel like the first two sections could provide more information about the importance and role of E3s to put the focus on them. 119, 120, 121, 122> The same concept is repeated 188>  Table 1.  Useful info but only BioID (not others) has been used in fusion to E3 substrate receptors (CRLs).  Table and discussion of other BirA derivatives does not fit title/topic of review. 235> Split-BioID is mentioned, but for no obvious reason related to E3 ligase substrates.  A Split-TurboID is the underlying basis of the SUMO-ID/UbL-ID method, and that method may match substrates to E3 ligases. 239> Section 4 is finally addressing the topic to which title/introduction refers, but the text/interpretation is very short and limited. 265> Authors mention AirID applied to CRBN and targeting protein degradation strategies. If this is mentioned, then a second report using BioID2-CRBN should also be mentioned (PMID 38975872). 269> “Wild-type BirA, has ” no comma necessary here. 269> This paragraph needs extensive editing.  They say that BioE3 identified substrates of TRAF6 and CHIP, but that was Ub-POD and they did not cite it.  Authors should check two references below that describe E-STUB and Ub-POD methods (very similar to BioE3) and be careful to match “E3 ligases tested” with each method to substrates identified. PMID: 38514884, 39121224. Authors should mention that BioE3, Ub-POD and E-STUB use variants of the AviTag with lower affinity for BirA, to enhance proximity dependence of the method. General comments: Some parts of the manuscript are repetitive, especially the introduction. Needs to be clearly stated that BioID-style approaches, when applied to E3 ligases, may identify substrates, but they should we validated orthogonally to distinguish true substrates from mere interactors.   BioE3/E-STUB/Ub-POD approaches are more likely to identify substrates since they depend on incorporation and biotinylation of Avitag-UB into substrates through short-range proximity with BirA, which occurs during E3 ligase-dependent substrate ubiquitination.  These methods are also amenable to tracking substrates/neosubstrates for particular E3 ligases using PROTACs and molecular glues (e.g. E-STUB paper and BioE3-CRBN. Other reviews focused on biotin-based methods for not only identifying E3 ligase substrates, but also other aspects of ubiquitin (total ubiquitomes, consequences of ubiquitination, etc.)  They should also be mentioned somewhere.  PMID: 38081789, 35181195. English is generally good, but some sentences could have better wording to accurately describe the concepts.

Author Response

1) A section on “future challenges” in this area would also be useful.

  Thank you for your suggestion. We added discussion about artifact induced by overexpression of biotin ligase-fused E3 ligases as a “future challenge” in Section 6 Conclusion as below (page 10, line 426 – page 10, line 436).

In most studies, biotin ligase-based proximity labeling has been performed by overexpressing the biotin ligase-fused POI. However, excess exogenous expression induces mislocalization and aberrant binding/biotinylation of the fusion protein, resulting in artifact detection [139]. To avoid such false candidates, the endogenous tagging of biotin ligases to genes encoding POI using genome editing would be helpful. A study using biotin ligase-tagged ubiquitin-specific peptidase 46 knock-in (KI) cells has identified several centrosomal proteins as interactors of ubiquitin-specific peptidase 46, suggesting the potential of biotin ligase-E3 ligase in KI cells to identify its substrates. In vivoapproaches using biotin ligase-fused E3 ligase of interest in KI animals would also be beneficial, especially for screening E3 ligase substrates that are ubiquitylated under specific physiological/pathological conditions.

2) 11-12> “Ubiquitylation is a post-translational modification that was originally identified as a molecular probe that leads labeled proteins…”  What is meant by “molecular probe”?

   Thank you for your pointing out. We changed the sentence as below to clarify the role of ubiquitylation (page 1, line 11 – page 1, line 12.)

   Ubiquitylation is a post-translational modification originally identified as the first step in protein degradation by the ubiquitin-proteasome system.

2) 39-40> “The 3D structure of the Ub-Ub molecule is differentiated by the position of the modulated lysine residue in the N-terminal Ub…”  This sentence is not clear and should be re-written; cannot understand what authors are trying to explain.

   As pointed out, we changed the sentence as below, and put it to the different place in the revised manuscript (page 1, line 39 – page 1, line 40).

   Different linkages between Ubs result in distinct chain conformations[8].

3) 72-77> This paragraph seems out of place, but “parkin” is mentioned, however not highlighted as an “E3 ubiquitin ligase”.

   As pointed out, we briefly mentioned that parkin is “an E3 ligase” as below (page 2, line 77 – page 2, line 80).

   In depolarized mitochondria, PTEN-induced putative kinase protein 1 (PINK1) accumulates on the outer membrane and phosphorylates Ub, which activates an E3 ligase Parkin and facilitates its recruitment to the mitochondria.

4) 84, 85> The concept of this sentence/ paragraph is repeated in the next one (line 88) The title of the work is “Identification of E3 ligase substrates using biotin ligase-based approaches”. I feel like the first two sections could provide more information about the importance and role of E3s to put the focus on them.

   As pointed out, we deleted the sentence at line 88 to avoid repetition. And as reviewer #2 and #3 suggested, we added more information to the section 1 and section 2 as below (page 3, line 88 – page 3, line 97; page 3, line 132 – page 4, line 166).

   Cell division is regulated by ubiquitylation. The E3 ligase anaphase-promoting complex/cyclosome is the main enzyme that targets essential mitotic factors for proteasomal degradation[26]. The anaphase-promoting complex/cyclosome ubiquitylates securin and cyclin B, which are important for regulating sister chromatid separation and mitotic exit, respectively, leading them to the Ub-proteasome pathway for their degradation. Non-proteolytic ubiquitylation plays an important role in cell division. The E3 ligase CUL3 engages the adaptor protein KLHL22, which ubiquitylates polo-like kinase 1 (PLK1), leading to the dissociation of PLK1 from kinetochores and ensuring a timely metaphase to anaphase transition[3,27]. The absence of PLK1 ubiquitylation blocks the dissociation of PLK1 from the kinetochore, resulting in prolonged mitotic arrest and cell death.

   2.2.   Ubiquitylation and neurodevelopmental disorders

Neurodevelopmental disorders are defined as “a group of conditions with onset in the developmental period, inducing deficits that produce impairments of functioning” in the New Diagnostic and Statistical Manual of Mental Disorders, 5th Edition (DSM-5)[53]. Approximately 40% of neurodevelopmental disorders are estimated to be caused by monogenic conditions, predominantly lesions in a single gene[31]. To date, many variants of E3 ligase-encoding genes have been identified in patients with neurodevelopmental disorders and proposed as causative factors for these disorders[31,54].

The UBE3A gene encoding the E6-AP HECT-type E3 ubiquitin ligase and loss-of-function mutation in the UBE3A has been shown to be a causative factor of Angelman syndrome, a neurodevelopmental disorder[31,55-57]. Intracellular localization and biochemical analyses revealed that 55% of the variants found in patients with Angelman syndrome induced mislocalization of UBE3A, and 29% of the variants induced a loss of E3 ligase activity[58]. Although the critical substrates of UBE3A are unknown, Arc has been identified as a substrate of UBE3A in neuronal cells[59]. Arc regulates the trafficking of alpha-amino-3-hydroxy-5-methyl-4-isoxazole-propionate-type glutamate receptors at synapses[60-62]. UBE3A ubiquitylates Arc and induces its degradation by the Ub-proteasome system; depletion of UBE3A increases Arc, followed by a reduction in cell surface expression of the alpha-amino-3-hydroxy-5-methyl-4-isoxazole-propionate receptor[58].

Genetic variants of the HECT E3 ligase NEDD4 subfamily have been proposed as causative factors of neurodevelopmental disorders[32]. The NEDD4 subfamily consists of 9 E3 ligases: NEDD4-1, NEDD4-2, ITCH, WWP1, WWP2, SMURF1, SMURF2, NEDL1, and NEDL2. These HECT E3 ligases play crucial roles in neurodevelopment[32]: NEDD4-1, SMURF1/2, WWP2, and ITCH1 in neuronal proliferation[63-68]; NEDD4-1/2, SMURF1, WWP1/2, and HECW2 in the migration and differentiation of neuronal cells[69-80]; and NEDD4-1/2 in the connections among neurons[81-85]. Genetic variants of NEDD4 have also been identified in patients with neurodevelopmental disorders. Genetic variants of NEDD4-2 have been identified in patients with periventricular nodular heterotopia, polymicrogyria, macrocephaly, cleft palate, and syndactyly[71]. Several studies have identified de novo mutations in the HECW gene in patients with neurodevelopmental diseases, including epilepsy, intellectual deficiency, and macrocephaly[77-80].

To date, pathogenic variants have been identified in at least 53 genes encoding E3 ligases in patients with neurodevelopmental disorders[54]. Further studies regarding the pathological relevance of E3 ligases in these disorders would be helpful for developing novel therapeutic approaches for these disorders.

5) 119, 120, 121, 122> The same concept is repeated

As pointed out, we edited the first paragraph of section2.3 (section 2.2 in the original manuscript) as below and delete a sentence from the second paragraph to avoid the repetition (page 4, line 168 – page 4, line 170).

Cancer cells modulate various cellular processes to actively and infinitely proliferate and evade anticancer systems, such as apoptosis and immunity, in the body. Cancer cells hijack ubiquitylation to induce this modulation[7].

6) 188> Table 1.  Useful info but only BioID (not others) has been used in fusion to E3 substrate receptors (CRLs).  Table and discussion of other BirA derivatives does not fit title/topic of review.

7) 235> Split-BioID is mentioned, but for no obvious reason related to E3 ligase substrates.  A Split-TurboID is the underlying basis of the SUMO-ID/UbL-ID method, and that method may match substrates to E3 ligases.

   Although some BirA derivatives and split-BioID might not have been used in fusion to E3 ligases, they have potential to be utilized for identification of substrates and interactors for E3 ligase. To clarify the statement, we added a sentence to the revised manuscript as below (page 7, line 296 – page 7, line 300).

   Various BioID derivatives and other biotin ligases have been developed to overcome limitations of BioID. Such derivatives and biotin ligases would be helpful for investigating the interactions between substrates/interactors and E3 ligases. In this subsection, we summarize improved BioIDs and other biotin ligases for proximity ligation, although some have not been used to identify substrates and interactors for E3 ligases.

8) 239> Section 4 is finally addressing the topic to which title/introduction refers, but the text/interpretation is very short and limited.

9) 265> Authors mention AirID applied to CRBN and targeting protein degradation strategies. If this is mentioned, then a second report using BioID2-CRBN should also be mentioned (PMID 38975872).

As suggested, we mentioned the second report using BioID2-CRBN (page 9, line 385 – page 9, line 391) and added some paragraphs to section 5 (section 4 in the original manuscript) in the revised manuscript as below (page 8, line 366 – page 8, line 371; page 9, line 396 – page 9, line 398; page 9, line 404 – page 9, line 409).

Our protein-protein interaction network analysis using the software Cytoscape (https://cytoscape.org) and a database String (using a plug-in for Cytoscape, stringApp) shows that RASSF3 and PPP1R15B interact with β-TrCP1 and β-TrCP2, respectively, TRIM9 and SUN2 directly interact with both β-TrCP1 and β-TrCP2, and STK3 and MAP1S indirectly interact with β-TrCP1, supporting their finding (Figure 3).

BioID2-based proximity labeling can also be used to investigate the toxic effects of IMiDs[132]. Screening of substrates/neo-substrates for CRBN/IMiD complexes using BioID2-fused CRBN identified non-muscle myosin heavy chain IIA (MYH9) as the neo-substrate. Although the protein level of MYH9 was not changed by treatment with IMiDs, ubiquitylated MYH9 levels increased in the presence of IMiDs. This study suggests the potential of BioID-based proximity labeling to identify neo-substrates of IMiDs and predict their side effects.

Additionally, these approaches used modified biotin acceptor peptides with low affinity for BirA to reduce the background biotinylation induced by the interaction between the acceptor peptides and BirA.

vinculin (VCL) and eukaryotic translation termination factor 1 (ETF1) (as neo-substrates for CRBN and an IMiD CC-885)[134], several kinases (as neo-substrates for CRBN and multikinase degraders SK-3-91, DB0646, SB1-G-187, and WH-10417-099)[134], SMRT/NcoR, MIER, CoREST, MiDAC, SIN3, and NuRD complexes [as neo-substrates for VHL and a dacinostat-based multi-histone deacetylase (HDAC) degrader XY-07-187][134], and HIF1a (as a substrate for VHL)[134].

10) 269> “Wild-type BirA, has ” no comma necessary here.

   As pointed out, we deleted the comma (page 9, line 392).

11) 269> This paragraph needs extensive editing.

They say that BioE3 identified substrates of TRAF6 and CHIP, but that was Ub-POD and they did not cite it. Authors should check two references below that describe E-STUB and Ub-POD methods (very similar to BioE3) and be careful to match “E3 ligases tested”with each method to substrates identified. PMID: 38514884, 39121224. Authors should mention that BioE3, Ub-POD and E-STUB use variants of the AviTag with lower affinity for BirA, to enhance proximity dependence of the method.

(general comments) BioE3/E-STUB/Ub-POD approaches are more likely to identify substrates since they depend on incorporation and biotinylation of Avitag-UB into substrates through short-range proximity with BirA, which occurs during E3 ligase-dependent substrate ubiquitination.  These methods are also amenable to tracking substrates/neosubstrates for particular E3 ligases using PROTACs and molecular glues (e.g. E-STUB paper and BioE3-CRBN. Other reviews focused on biotin-based methods for not only identifying E3 ligase substrates, but also other aspects of ubiquitin (total ubiquitomes, consequences of ubiquitination, etc.)  They should also be mentioned somewhere.  PMID: 38081789, 35181195.

   Sorry for the incorrect citation. We carefully checked the references, corrected and added citations properly, mentioned E-STUB and Ub-POD, and edited the paragraph as below  (page 9, line 392 – page 9, line 409).

   Wild-type BirA has also been utilized to identify E3 ligase substrates. In this approach, known as BioE3, E-STAB, or Ub-POD, Ub tagged with a biotin acceptor peptide such as AviTag is used as an acceptor for biotinylation by BirA-tagged E3 ligases[133-135]. Since BirA biotinylates its target sequence in a specific manner, ubiquitylated proteins should be preferentially identified using this approach. Additionally, these approaches used modified biotin acceptor peptides with low affinity for BirA to reduce the background biotinylation induced by the interaction between the acceptor peptides and BirA. The wild-type BirA-based approaches have been used to identify various candidate substrates for E3 ligases, such as many components of the replication fork and proteins with helicase activity (as substrates for RNF4) [133], ESCRT proteins signal transducing adapter molecule (STAM) [135] and hepatocyte growth factor-regulated tyrosine kinase substrate (HGS) [135] and, selective autophagy receptor Toll-interacting protein (as substrates for TRAF6), annexin (as a substrate for CHIP)[135], vinculin (VCL) and eukaryotic translation termination factor 1 (ETF1) (as neo-substrates for CRBN and an IMiD CC-885)[134], several kinases (as neo-substrates for CRBN and multikinase degraders SK-3-91, DB0646, SB1-G-187, and WH-10417-099)[134], SMRT/NcoR, MIER, CoREST, MiDAC, SIN3, and NuRD complexes [as neo-substrates for VHL and a dacinostat-based multi-histone deacetylase (HDAC) degrader XY-07-187][134], and HIF1a (as a substrate for VHL)[134].

12) General comments: Some parts of the manuscript are repetitive, especially the introduction.

   Thank you for pointing out. We carefully revised our manuscript.

13) Needs to be clearly stated that BioID-style approaches, when applied to E3 ligases, may identify substrates, but they should we validated orthogonally to distinguish true substrates from mere interactors.

   As suggested, we mentioned the necessity of validation of candidate substrates identified by biotin-ligase-based proximity labeling as below (page 10, line 423 – page 10, line 425).

Although biotin ligase-based proximity labeling is helpful for identifying substrates for E3 ligases, validation using other methods, such as immunoprecipitation, is required to confirm that the identified candidates are true substrates.

14) Other reviews focused on biotin-based methods for not only identifying E3 ligase substrates, but also other aspects of ubiquitin (total ubiquitomes, consequences of ubiquitination, etc.)  They should also be mentioned somewhere.  PMID: 38081789, 35181195.

   Thank you for your suggestion. As the main focus of the manuscript is E3 ligase substrates, we briefly mentioned a potential of biotin ligase-based proximity labeling for analysis on other aspects of ubiquitylation in conclusions as below (page 10, line 442 – page 10, line 445).

In addition to E3 ligase substrates/interactors, other aspects of ubiquitylation, such as the ubiquitome, have been analyzed using biotin ligases[141-144]. Several questions regarding ubiquitylation could be resolved using biotin ligase-based proximity labeling approaches.

Reviewer 3 Report

Comments and Suggestions for Authors

Summary
The authors provide an insightful review of the ubiquitylation process. However, references must be formatted according to MDPI’s Instructions for Authors. Additionally, the manuscript would benefit from a dedicated section on the role of protein ubiquitylation in neurodevelopmental disorders. A more in-depth meta-analysis using online databases is also recommended.

Title
Ensure the title adheres to MDPI’s Instructions for Authors by capitalizing each relevant word.

Introduction

Figure 1 should clearly indicate the original source from which it has been adapted; The figure’s description should provide a brief but informative overview of the ubiquitylation process.

Introduction section should be concluded with a compelling statement that highlights the aim of this review while effectively transitioning into the subsequent sections.

Section 2

  • Beyond cancer and neurodegenerative diseases, recent research highlights the involvement of ubiquitylation defects in neurodevelopmental disorders (NDDs), including autism and intellectual disability.
  • Define NDDs according to DSM-5 (add appropriate reference).
  • Discuss specific genes and pathways related to ubiquitylation that have been linked to NDDs. The following references may be helpful: https://doi.org/10.1038/s41598-024-66475-2;  https://doi.org/10.3389/fnmol.2021.733012; https://doi.org/10.3390/ijms23073882
  • A new section on ubiquitylation as a therapeutic target should be included. Consider incorporating insights from: https://doi.org/10.1016/j.jbc.2024.107264; https://doi.org/10.1098/rsob.150018

Table 1
Ensure Table 1 is formatted according to MDPI’s template.

Subsection 3.2

  • Have you cross-checked your approach with the list of interactions and interactors available in the BioGRID database?
  • Consider incorporating specific ubiquitylation pathways and protein interactions identified using Cytoscape (by querying a list of proteins) and NDEx databases:
    • https://www.ndexbio.org/index.html#/

Author Response

1) However, references must be formatted according to MDPI’s Instructions for Authors.

   Thank you for pointing out. We formatted references according to MDPI’s Instructions for Authors.

2) Title

Ensure the title adheres to MDPI’s Instructions for Authors by capitalizing each relevant word.

   We appreciate your pointing out. We capitalized each relevant word of the title (page 1, line 2 – page 1, line 3).

2) Introduction

Figure 1 should clearly indicate the original source from which it has been adapted; The figure’s description should provide a brief but informative overview of the ubiquitylation process.

   As reviewer #3 pointed out, we added a citation (Sun, Cell and Bioscience, 2022) to Figure 1 and description about overview of the ubiquitylation process to the legend as below (page 2, line 46 – page 2, line 49).

    1) Ub is activated and crosslinked to E1. 2) Activated Ub is transferred from E1 to E2. 3) E3 binds both Ub-linked E2 and a substrate, then catalyzes the transfer of Ub from E2 to a lysine residue of the substrate or Ub conjugated to the substrate.

3) Introduction section should be concluded with a compelling statement that highlights the aim of this review while effectively transitioning into the subsequent sections.

   As reviewer #3 suggested, we added a highlight of the aim of this review to the revised manuscript as below (page 3, line 100 – page 3, line 102).

Therefore, the identification of substrates for each E3 ligase is important for understanding how cellular processes are maintained and how abnormalities in ubiquitylation cause various diseases.

4) (Summary) Additionally, the manuscript would benefit from a dedicated section on the role of protein ubiquitylation in neurodevelopmental disorders.

Section 2

  • Beyond cancer and neurodegenerative diseases, recent research highlights the involvement of ubiquitylation defects in neurodevelopmental disorders (NDDs), including autism and intellectual disability.
  • Define NDDs according to DSM-5 (add appropriate reference).
  • Discuss specific genes and pathways related to ubiquitylation that have been linked to NDDs. The following references may be helpful: https://doi.org/10.1038/s41598-024-66475-2; https://doi.org/10.3389/fnmol.2021.733012; https://doi.org/10.3390/ijms23073882

   As suggested, we added neurodevelopmental diseases as diseases which pathogenesis is associated with abnormalities in ubiquitylation (page 3, line 105) and a subsection entitled “Ubiquitylation and neurodevelopmental disorders” to section 2 as below (page 3, line 132 – page 4, line 166).

   2.2.   Ubiquitylation and neurodevelopmental disorders

Neurodevelopmental disorders are defined as “a group of conditions with onset in the developmental period, inducing deficits that produce impairments of functioning” in the New Diagnostic and Statistical Manual of Mental Disorders, 5th Edition (DSM-5)[53]. Approximately 40% of neurodevelopmental disorders are estimated to be caused by monogenic conditions, predominantly lesions in a single gene[31]. To date, many variants of E3 ligase-encoding genes have been identified in patients with neurodevelopmental disorders and proposed as causative factors for these disorders[31,54].

The UBE3A gene encoding the E6-AP HECT-type E3 ubiquitin ligase and loss-of-function mutation in the UBE3A has been shown to be a causative factor of Angelman syndrome, a neurodevelopmental disorder[31,55-57]. Intracellular localization and biochemical analyses revealed that 55% of the variants found in patients with Angelman syndrome induced mislocalization of UBE3A, and 29% of the variants induced a loss of E3 ligase activity[58]. Although the critical substrates of UBE3A are unknown, Arc has been identified as a substrate of UBE3A in neuronal cells[59]. Arc regulates the trafficking of alpha-amino-3-hydroxy-5-methyl-4-isoxazole-propionate-type glutamate receptors at synapses[60-62]. UBE3A ubiquitylates Arc and induces its degradation by the Ub-proteasome system; depletion of UBE3A increases Arc, followed by a reduction in cell surface expression of the alpha-amino-3-hydroxy-5-methyl-4-isoxazole-propionate receptor[58].

Genetic variants of the HECT E3 ligase NEDD4 subfamily have been proposed as causative factors of neurodevelopmental disorders[32]. The NEDD4 subfamily consists of 9 E3 ligases: NEDD4-1, NEDD4-2, ITCH, WWP1, WWP2, SMURF1, SMURF2, NEDL1, and NEDL2. These HECT E3 ligases play crucial roles in neurodevelopment[32]: NEDD4-1, SMURF1/2, WWP2, and ITCH1 in neuronal proliferation[63-68]; NEDD4-1/2, SMURF1, WWP1/2, and HECW2 in the migration and differentiation of neuronal cells[69-80]; and NEDD4-1/2 in the connections among neurons[81-85]. Genetic variants of NEDD4 have also been identified in patients with neurodevelopmental disorders. Genetic variants of NEDD4-2 have been identified in patients with periventricular nodular heterotopia, polymicrogyria, macrocephaly, cleft palate, and syndactyly[71]. Several studies have identified de novo mutations in the HECW gene in patients with neurodevelopmental diseases, including epilepsy, intellectual deficiency, and macrocephaly[77-80].

To date, pathogenic variants have been identified in at least 53 genes encoding E3 ligases in patients with neurodevelopmental disorders[54]. Further studies regarding the pathological relevance of E3 ligases in these disorders would be helpful for developing novel therapeutic approaches for these disorders.

5) •        A new section on ubiquitylation as a therapeutic target should be included. Consider incorporating insights from: https://doi.org/10.1016/j.jbc.2024.107264; https://doi.org/10.1098/rsob.150018

   Thank you for your suggestion. We added a new section entitled “Ubiquitylation as a therapeutic target” to the revised manuscript as below (page 5, line 196 – page 6, line 235).

  1. Ubiquitylation as a therapeutic target

Protein ubiquitylation is an attractive target for the development of medicines. Various therapeutic approaches using ubiquitylation have been proposed.

Thalidomide is a sedative with teratogenic side effects; however, its derivatives, including lenalidomide and pomalidomide, are also relatively safe immunomodulatory drugs (IMiDs) for patients with cancer, except for pregnant women[93]. IMiDs have been shown to interact with cereblon (CRBN), an adapter protein of E3 ligase, recruiting proteins such as the Ikaros family zinc finger protein 1 (IKZF1) and IKZF3, which play central roles in the biology of B and T cells, as neo-substrates[94-100]. This interaction facilitates the proteasome-mediated degradation of IKZF1 and IKZF3. This unique characteristic of IMiDs is related to their anti-multiple-myeloma activities.

CC-885 was identified as a CRBN ligand with unique activities from a library of thalidomide analogues[101]. CC-885 binds to CRBN and GSPT1 (eRF3a), a translation termination factor that induces ubiquitination and degradation by the Ub-proteasome, as well as IKZF1 and IKZF3. CC-885 exhibits anti-tumor effects through the degradation of GSPT1.

Indisulam (E7070) is a sulphonamide anti-tumor drug[93]. Indisulam disrupts and reduces the levels of cyclin A, cyclin B, CDK2, and CDC2 through mechanisms that depend on p21 and p53, affecting multiple checkpoints during the G1 and G2 phases of the cell cycle[102]. A study on indisulam resistance, which is caused by mutations in RBM39, an RNA-binding nuclear protein, revealed that indisulam binds to both RBM39 and a component of the CUL4 E3 ligase complex, DCAF15, inducing polyubiquitination and degradation of RBM39[103]. The degradation of RBM39 induced by indisulam treatment leads to aberrant pre-mRNA splicing, which, in turn, induces cell death in cancer cell lines that highly express RBM39.

Recently, a promising therapeutic approach, proteolysis-targeting chimera (PROTAC) technology, was introduced to promote the degradation of target proteins via the Ub-proteasome system[104]. PROTAC is a dual-binder comprising an E3 ligase-binding ligand and a protein of interest (POI )-binding ligand. PROTAC recruits a POI to its target E3 ligase, followed by ubiquitylation and degradation of the POI.

Vepdegestrant (ARV-471) is a PROTAC composed of a lasofoxifene-based estro-gen receptor-targeting moiety linked to a derivative of lenalidomide, a clinical IMiD that binds to CRBN[105]. Vepdegestrant induces degradation of the estrogen receptor through the Ub-proteasome system via polyubiquitylation by CRBN-containing E3 ligase complexes. A preclinical study using MCF7 orthotopic xenograft models and a patient-derived xenograft breast cancer model has demonstrated that vepdegestrant has potent anti-tumor activity against estrogen receptor-positive tumors[105]. A Phase III clinical trial is currently in progress[106].

Dual-binding molecules targeting E3 ligases and pathogenic proteins/disease-specific proteins are promising therapeutic drugs for treating intractable cancers and various diseases that are induced by pathogenic proteins or expressed disease-specific proteins.

6) Table 1

Ensure Table 1 is formatted according to MDPI’s template.

   As suggested, we formatted Table1 according to MDPI's template (page 7, line 280 – page 7, line 294).

7) Subsection 3.2

  • Have you cross-checked your approach with the list of interactions and interactors available in the BioGRID database?

   Thank you for your suggestion. As suggested, we cross-checked the approach using BioID2 with the list of interactions and interactiors available in the BioGRID database and mentioned it as below (page 7, line 312 – page 7, line 314).

Using BioGRID, we confirmed that, in the top10 hits identified using BioID2-fused nucleoporin component Nup43, two of the candidates have been shown their interaction with Nup43 by other methods[126,127].

8) •        Consider incorporating specific ubiquitylation pathways and protein interactions identified using Cytoscape (by querying a list of proteins) and NDEx databases:

  • https://www.ndexbio.org/index.html#/

   As suggested, we added protein-protein interaction of β-TrCP1 (BTRC) and β-TrCP2 (FBXW11) with their substrates identified in Coyaud, E. et al, Mol Cell Proteomics 2015, 14 (7), 1781-1795 as figure 3, and mentioned the interaction as below (page 8, line 366 – page 8, line 371).

   Our protein-protein interaction network analysis using the software Cytoscape (https://cytoscape.org) and a database String (using a plug-in for Cytoscape, stringApp) shows that RASSF3 and PPP1R15B interact with β-TrCP1 and β-TrCP2, respectively, TRIM9 and SUN2 directly interact with both β-TrCP1 and β-TrCP2, and STK3 and MAP1S indirectly interact with β-TrCP1, supporting their finding (Figure 3).

Round 2

Reviewer 2 Report

Comments and Suggestions for Authors

Reviewer thanks the authors for careful revision and point-by-point response.

In response, authors propose new text: "Although biotin ligase-based proximity labeling is helpful for identifying substrates for E3 ligases, validation using other methods, such as immunoprecipitation, is required to confirm that the identified candidates are true substrates."

Not sure that I agree with this statement, since ligase/substrate interaction may be too weak or too transient (or both) to be captured convincingly in a co-immunoprecipitation experiment.

Also, authors should revise again manuscript and, when using "biotin ligase" in a sentence or paragraph, distinguish between "biotin ligase" (i.e. wild-type BirA) and "promiscuous biotin ligase" (BioID, BioID2, TurboID, miniTurbo, AirID, UltraID etc...).   This will help reader to understand and keep track of which applications are using BirA or Promiscous BirA derivatives.

With these minor revisions, the review is acceptable and should be published.

Author Response

Responses to the comments raised by Reviewer #2

1) In response, authors propose new text: "Although biotin ligase-based proximity labeling is helpful for identifying substrates for E3 ligases, validation using other methods, such as immunoprecipitation, is required to confirm that the identified candidates are true substrates."

Not sure that I agree with this statement, since ligase/substrate interaction may be too weak or too transient (or both) to be captured convincingly in a co-immunoprecipitation experiment.

  Thank you for your suggestion. We added immunoprecipitation using both substrate candidates- and E3 ligase of interest-overexpressing cells and analysis of ubiquitylation of the candidates in E3 ligase of interest-knockout/knockdown cells as validation methods to the sentence as below (page 10, line 425 – page 10, line 429).

Although biotin ligase-based proximity labeling is helpful for identifying substrates for E3 ligases, validation using other methods, such as immunoprecipitation using cells expressing both substrate candidates and E3 ligase of interest, is required to confirm that the identified candidates are true substrates. Defects of ubiquitylation of the candidates should be also observed in E3 ligase of interest-knockout/knockdown cells.

2) Also, authors should revise again manuscript and, when using "biotin ligase" in a sentence or paragraph, distinguish between "biotin ligase" (i.e. wild-type BirA) and "promiscuous biotin ligase" (BioID, BioID2, TurboID, miniTurbo, AirID, UltraID etc...).   This will help reader to understand and keep track of which applications are using BirA or Promiscous BirA derivatives.

   Thank you for your suggestion. We clarified that wild-type BirA does not diffuse BioAMP to the surrounding environment as below (page 9, line 392 – page 9, line 394.)

   Wild-type BirA has also been utilized to identify E3 ligase substrates. Unlike promiscuous biotin ligases such as BioID, wild-type BirA does not diffuse bioAMP to the surrounding environment.

Reviewer 3 Report

Comments and Suggestions for Authors

Authors thoroughly addressed all the reviewer's comments

Author Response

Thank you for your comment.